# Blind Super-Resolution Kernel Estimation using an Internal-GAN

**Sefi Bell-Kligler**      **Assaf Shocher**      **Michal Irani**
Dept. of Computer Science and Applied Math
The Weizmann Institute of Science, Israel
Project website: http://www.wisdom.weizmann.ac.il/~vision/kernelgan

## Abstract

Super resolution (SR) methods typically assume that the low-resolution (LR) image was downscaled from the unknown high-resolution (HR) image by a fixed 'ideal' downscaling kernel (e.g. Bicubic downscaling). However, *this is rarely the case in real LR images*, in contrast to synthetically generated SR datasets. When the assumed downscaling kernel deviates from the true one, the performance of SR methods significantly deteriorates. This gave rise to *Blind-SR* – namely, SR when the downscaling kernel ("SR-kernel") is unknown. It was further shown that the true SR-kernel is the one that maximizes the recurrence of patches across scales of the LR image. In this paper we show how this powerful cross-scale recurrence property can be realized using *Deep Internal Learning*. We introduce "KernelGAN", an image-specific *Internal-GAN* [29], which trains solely on the LR test image at test time, and learns its internal distribution of patches. Its *Generator* is trained to produce a downscaled version of the LR test image, such that its *Discriminator* cannot distinguish between the patch distribution of the downscaled image, and the patch distribution of the original LR image. *The Generator, once trained, constitutes the downscaling operation with the correct image-specific SR-kernel*. KernelGAN is fully unsupervised, requires no training data other than the input image itself, and leads to state-of-the-art results in Blind-SR when plugged into existing SR algorithms. [1]

## 1   Introduction

The basic model of SR assumes that the low-resolution input image $I_{LR}$ is the result of down-scaling a high-resolution image $I_{HR}$ by a scaling factor $s$ using some kernel $k_s$ (the "SR kernel"), namely:

$$I_{LR} = (I_{HR} * k_s) \downarrow_s \qquad (1)$$

The goal is to recover $I_{HR}$ given $I_{LR}$. This problem is ill-posed even when the SR-Kernel is assumed known (an assumption made by most SR methods – older [8, 32, 7] and more recent [5, 20, 19, 21, 38, 35, 12]). A boost in SR performance was achieved in the past few years introducing Deep-Learning based methods [5, 20, 19, 21, 38, 35, 12]. However, since most SR methods train on synthetically downscaled images, they implicitly rely on the SR-kernel $k_s$ being fixed and 'ideal' (usually a Bicubic downscaling kernel with antialiasing– MATLAB's default imresize command). **Real LR images, however, rarely obey this assumption**. This results in poor SR performance by state-of-the-art (SotA) methods when applied to real or 'non-ideal' LR images (see Fig. 1a).

The SR kernel of real LR images is influenced by the sensor optics as well as by tiny camera motion of the hand-held camera, resulting in a *different non-ideal SR-kernel for each LR image*, even if taken by the same sensor. It was shown in [26] that the effect of using an incorrect SR-kernel is of greater

**(a)** **Comparison to SotA SR methods (SR×4):**
Since they train on *'ideal' LR images*, they perform poorly on *real non-ideal LR images*.

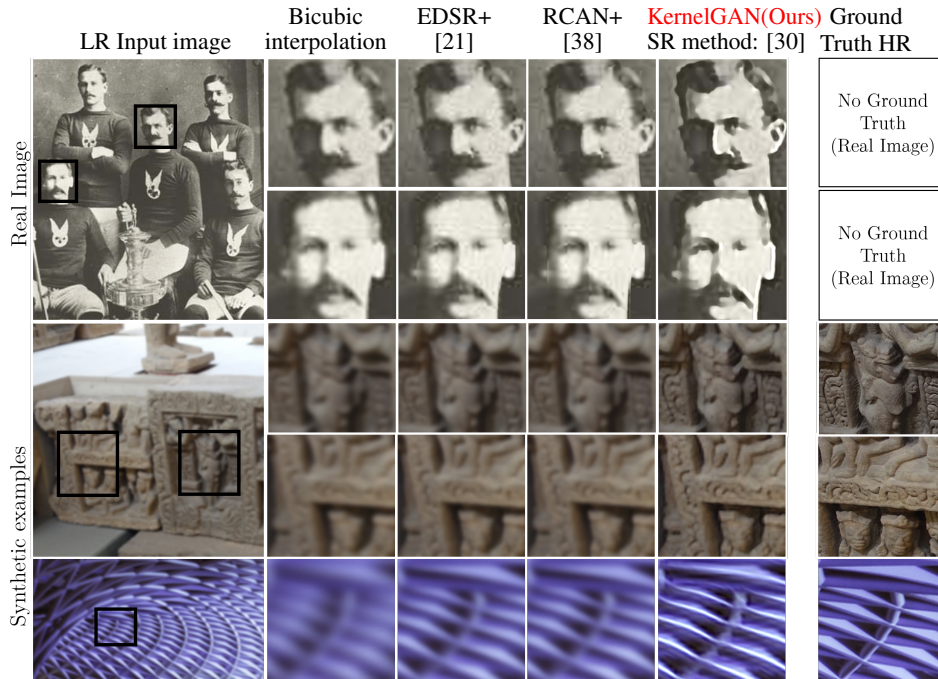

**(b)** **Comparison to SotA *Blind*-SR methods (SR×4):**

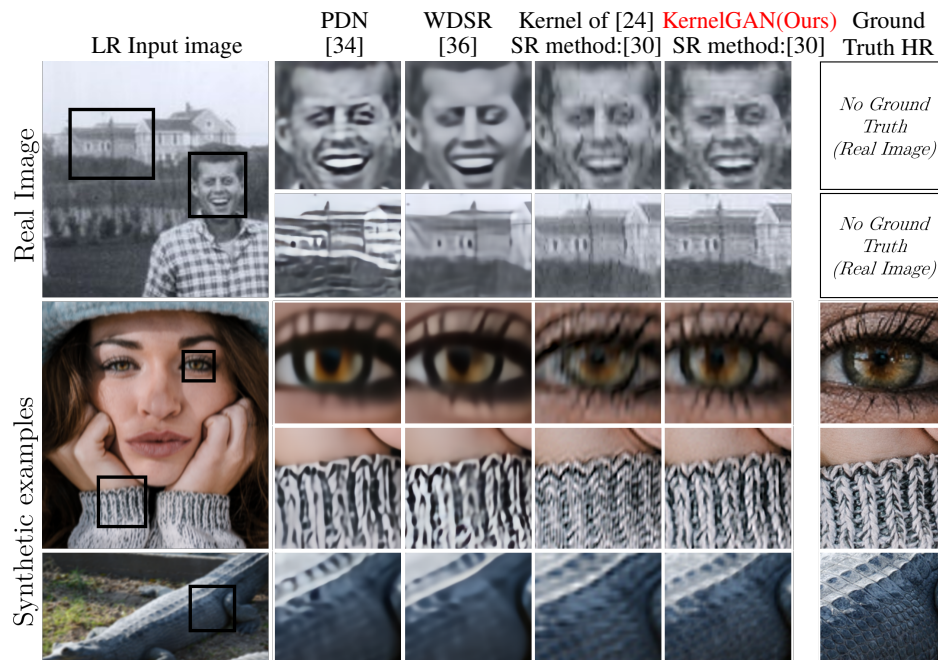

Figure 1: **SR×4 on real 'non-ideal' LR images** *(downloaded from the internet, or downscaled by an unknown kernel). Full images and additional examples in supplementary material (*please zoom in on screen).

influence on the SR performance than any choice of an image prior. This observation was later shown to hold also for deep-learning based priors [30]. The importance of the SR-kernel accuracy was further analyzed empirically in [27].

The problem of SR with an unknown kernel is known as *Blind SR*. Some Blind-SR methods [33, 17, 14, 13, 3] were introduced prior to the deep learning era. Few Deep Blind-SR methods [34, 36] were recently presented in the NTIRE'2018 SR challenge [31]. These methods do not explicitly calculate the SR-kernel, but propose SR networks that are robust to variations in the downscaling kernel. A work concurrent to ours, IKC [10], performs Blind-SR by alternating between the kernel estimation and the SR image reconstruction. A different family of recent Deep SR methods [30, 37], while not explicitly Blind-SR, allow to input a different *image-specific* (pre-estimated) SR-kernel along with the LR input image at test time. Note that *SotA (non-blind) SR methods cannot make any use of an image-specific SR kernel at test time* (even if known/provided), since it is different from the fixed downscaling kernel they trained on. These methods thus produce poor SR results in realistic scenarios – see Fig. 1a (in contrast to their *excellent* performance on synthetically generated LR images).

The recurrence of small image patches (5x5, 7x7) across scales of a single image, was shown to be a very strong property of natural images [8, 39]. Michaeli & Irani [24] exploited this recurrence property to estimate the unknown SR-kernel directly from the LR input image. An important observation they made was that **the correct SR-kernel** is also the downscaling kernel which **maximizes the similarity of patches across scales of the LR image.** Based on their observation, they proposed a nearest-neighbor patch based method to estimate the kernel. However, their method tends to fail for SR scale factors larger than ×2, and has a very long runtime.

This *internal* cross-scale recurrence property is very powerful, since it is *image-specific* and unsupervised (requires no prior examples). In this paper we show how this property can be combined with the power of Deep-Learning, to obtain the best of both worlds – **unsupervised** **SotA SR-kernel estimation, with SotA Blind-SR results**. We build upon the recent success of *Deep Internal Learning* [30] (training an *image-specific* CNN on examples extracted directly from the test image), and in particular on *Internal-GAN* [29] – a self-supervised GAN which learns the image-specific distribution of patches.

More specifically, we introduce "KernelGAN" – an image-specific *Internal-GAN*, which *estimates the SR kernel that best preserves the **distribution of patches** across scales of the LR image*. Its *Generator* is trained to produce a downscaled version of the LR test image, such that its *Discriminator* cannot distinguish between the patch distribution of the downscaled image, and the patch distribution of the original LR image. In other words, $G$ trains to fool $D$ to believe that all the patches of the downscaled image were actually taken from the original one. ***The Generator, once trained, constitutes the downscaling operation with the correct image-specific SR-kernel***. KernelGAN is fully unsupervised, requires no training data other than the input image itself, and leads to state-of-the-art results in Blind-SR when plugged into existing SR algorithms.

Since downscaling by the SR-kernel is a *linear* operation applied to the LR image (convolution and subsampling – see Eq. 1), our Generator (as opposed to the Discriminator) is a *linear network* (without non-linear activations). At first glance, it may seem that a single-strided convolution layer (which stems from Eq. 1) should suffice as a Generator. Interestingly, we found that using a *deep linear network* is dramatically superior to a single-strided one. This is consistent with recent findings in theoretical deep-learning [28, 2, 18, 11], where deep linear networks were shown to have optimization advantages over a single layer network for linear regression models. This is further elaborated in Sec. 4.1.

Our contributions are therefore several-fold:

- This is the first dataset-invariant deep-learning method to estimate the unknown SR kernel (a critical step for true SR of *real* LR images). KernelGAN is fully unsupervised and requires no training data other than the input image itself, hence enables true SR in "the wild".
- KernelGAN leads to state-of-the-art results in Blind-SR when plugged into existing SR algorithms.
- To the best of our knowledge, this is the first practical use for deep *linear* networks (so far used mostly for theoretical analysis), with demonstrated practical advantages.

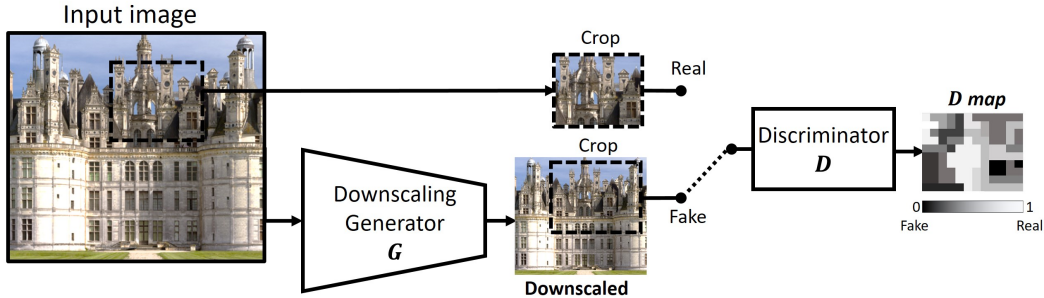

Figure 2: **KernelGAN**: *The patch GAN trains on patches of a single input image (real). $D$ tries to distinguish real patches from those generated by $G$ (fake). $G$ learns to downscale $X2$ the image while fooling $D$ i.e. maintaining the same distribution of patches. Both networks are fully convolutional, which in the case of images implies that each pixel in the output is a result of a specific receptive field (i.e. patch) in the input.*

## 2   Overview of the Approach

Given only the input image, our goal is to find its underlying image-specific SR kernel. We seek the kernel that best preserves the distribution of patches across scales of the LR image. More specifically, we aim to "generate" a downscaled image (e.g. by a factor of 2) such that its patch distribution is as close as possible to that of the LR image.

By matching distributions rather than individual patches, we can leverage recent advances in distribution modeling using Generative Adversarial Networks (GANs) [9]. GANs can be understood as a tool for distribution matching [9]. A GAN typically learns a **distribution of images** in a large image dataset. It maps examples from a source distribution, $p_x$, to examples indistinguisable from a target distribution, $p_y$: $G : x \rightarrow y$ with $x \sim p_x$, and $G(x) \sim p_y$. An internal GAN [29] trains on a single input image and learns its unique internal **distribution of patches**.

Inspired by InGAN [29], KernelGAN is also an image specific GAN that *trains on a single input image.* It consists of a downscaling generator ($G$) and a discriminator ($D$) as depicted in Fig. 2. Both G and D are fully-convolutional, which implies the network is applied to patches rather than the whole image, as in [16]. Given an input image $I_{LR}$, $G$ learns to downscale it, such that, for $D$, at the *patch level*, it is indistinguishable from the input image $I_{LR}$.

$D$ trains to output a heat map, referred to as D-map (see fig. 2) indicating for each pixel, how likely is its surrounding patch to be drawn from the original patch-distribution. It alternately trains on real examples (crops from the input image) and fake ones (crops from $G$'s output). The loss is the pixel-wise MSE difference between the output D-map and the label map. The labels for training $D$ are a map of all ones for crops extracted from the original LR image, and a map of all zeros for crops extracted from the downscaled image.

We adopt a variant of the LSGAN [23] with the L1 Norm, and define the objective of KernelGAN as:

$$G^*(I_{LR}) = \underset{G}{\mathrm{argmin}} \max_{D} \left\{ \mathbb{E}_{x \sim \text{patches}(I_{LR})}[|D(x) - 1| + |D(G(x))|] + \mathcal{R} \right\} \quad (2)$$

where $\mathcal{R}$ is the regularization term on the downscaling SR-kernel resulting from the generator $G$ (see more details in Sec. 4.2). Once converged, **the generator $G^*$ constitutes, implicitly, the ideal SR downscaling function for the specific LR image**.

The GAN trains for 3,000 iterations, alternating single optimization steps of $G$ and $D$, with the ADAM optimizer ($\beta_1 = 0.5, \beta_2 = 0.999$). Learning rate is $2e^{-4}$, decaying $\times 0.1$ every 750 iters.

## 3   Discriminator

The goal of the discriminator $D$ is to learn the distribution of patches of the input image $I_{LR}$ and discriminate between real patches belonging to this distribution and fake patches generated by $G$. $D$'s real examples are crops from $I_{LR}$, while fake examples are crops currently outputed by $G$.

We use a fully-convolutional patch discriminator, as introduced in [16], applied to learn the patch distribution of a single image as in [29]. To discriminate small image patches, we use no pooling

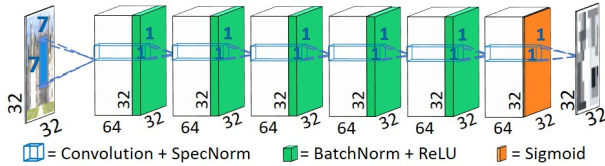

Figure 3: **Fully Convolutional Patch Discriminator**: *A 7×7 convolution filter followed by six 1×1 convolutions, including Spectral normalization [25], Batch normalization [15], ReLU and a Sigmoid activations. An input crop of 32×32 results with a 32×32 map∈ [0,1].*

= Convolution + SpecNorm   = BatchNorm + ReLU   = Sigmoid

or strides, thus achieving a small receptive field of a 7×7 patch. In this setting, $D$ is implicitly applied to each patch separately and produces a heat-map ($D$-map) where each location in the map corresponds to a patch from the input. The labels for the discriminator are maps (matrices of real/fake, i.e. 1/0 labels, respectively) of the same size as the input to the discriminator. Each pixel in $D$-map indicates how likely is its surrounding patch to be drawn from the learned patch distribution. See Fig. 3 for architecture details of the discriminator.

# 4 Deep Linear Generator = The downscaling SR-Kernel

## 4.1 Deep Linear Generator

The generator $G$ constitutes the downscaling model. Since downscaling by convolution and subsampling is a *linear* transform (Fig. 1), we use a *linear* Generator (without any non-linear activations). We refer to the model of downscaling from Eq. 1. In principle, the expressiveness of a single strided convolutional layer should cover all possible downscaling methods captured by Eq. 1. However, we *empirically* found that such architecture does not converge to the correct solution (see 6). We attribute this behavior to the following *conjecture*: A generator consisting of a single layer has exactly one set of parameters for the correct solution (the set of weights that make the ground-truth kernel). This means that there is only one point on the optimization surface that is acceptable as a solution and it is the global minimum. Achieving such a global minimum is easy when the loss function is convex (as in linear regression). But in our case the overall loss function is a non-linear neural network-$D$ which is highly non-convex. In such case, the probability for getting from some random initial state to the global minimum using gradient based methods is negligible. In contrast to a single layer, standard *(*deep) neural networks are assumed to converge from a random initialization due to the fact that there are many good local minima and negligibly few bad local minima [6, 18]. Hence, for a problem that by definition has one valid solution, optimizing a single layer generator is impossible.

A non-linear generator would not be suitable either. Without an explicit constraint on $G$ to be linear, it will generate physically unwanted solutions for the optimization objective. Such solutions include generating any image that contains valid patches but has no downscaling relation (Eq. 1). One example is generating a tile of just several patches from the input. This would be a solution that complies with eq. 2 but is unwanted.

This conjecture motivated us to use **deep linear networks**. These are networks consisting of a sequence of linear layers with *no activations*, and are used for theoretical analysis [28, 2, 18, 11]. The expressiveness of a deep linear network is exactly as the one of a single linear layer (i.e. Linear regression), however, its optimization has several different aspects. While it can be convex with respect to the input, the loss would never be convex with respect to the weights (assuming more than one layer). In fact, linear networks have infinitely many equally valued global minima. Any choice of network parameters matching to one of these minima would be equivalent to any other minimum point-they are just different factorizations of the same matrix. Motivated by these observations, we employ a deep linear generator. By that we allow infinitely many valid solutions to our optimization objective, all equivalent to the same linear solution. Furthermore, it was shown by [2], that gradient-based optimization is faster for deeper linear networks than shallow ones.

Fig. 4 depicts the architecture of $G$. We use 5 hidden convolutional layers with 64 channels each. The first 3 filters are 7×7, 5×5, 3×3 and the rest are 1×1. This makes a receptive field of 13 ×13 (allowing for a 13 × 13 SR kernel). This setting of filters takes into account the effective receptive field [22]; maintaining the same receptive field while having filters bigger than 1×1 following the first layer encourages the center of the kernel to have higher values. To provide a reasonable initial starting point, the generator's output is constrained to be similar to an ideal downscaling (e.g. bicubic) of the input, for the first iterations. Once satisfied, this constraint is discarded.

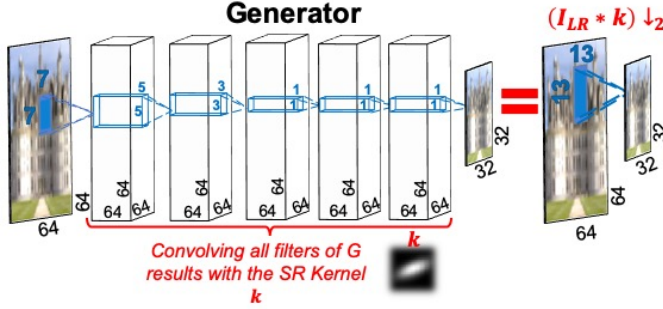

Figure 4: **Deep linear G**: *A 6 layer convolutional network without non-linear activations.*
*The deep linear network on the left has equal expressiveness power as the single strided convolution downscaling model, on the right.*

## 4.2 Extracting the explicit kernel

At any iteration during the training, the generator $G$ constitutes the *currently* estimated downscaling operation for the specific LR input image. The image-specific kernel is **implicitly** captured by the trained weights of $G$. However, there are two reasons to **explicitly** extract the kernel from $G$. First, we are interested in having a compact downscaling convolution kernel rather than a downscaling network. This step is trivial; convolving all the filters of $G$ sequentially with sride 1 results in the SR-kernel $k$ (see Fig. 4). When extracted, the kernel is just a small array that can be supplied to SR algorithms. The second reason for explicitly extracting the kernel is to allow applying explicit and physically-meaningful priors on the kernel. This is the goal of the regularization term $\mathcal{R}$ in Eq.2 which decreases the hypotheses space to a sub-set of the plausible kernels, that obey certain restrictions. Such restrictions are that the kernel would sum up to 1 and be centered so it will not shift the image. The regularization also ameliorates faulty tendencies of the optimization process to produce kernels that are too spread out and smooth. However, it is not enough to extract the kernel, this extraction must be differentiable so that the regularization losses can be back-propagated through it. At each iteration, we apply a differentiable action of calculating the kernel (convolving all the filters of $G$ sequentially with stride-1). The regularization loss term $\mathcal{R}$ is then applied and included in the optimization objective.

The regularization term in our objective in eq. 2 is the following:

$$\mathcal{R} = \alpha \mathcal{L}_{sum\_to\_1} + \beta \mathcal{L}_{boundaries} + \gamma \mathcal{L}_{sparse} + \delta \mathcal{L}_{center} \tag{3}$$

where $\alpha = 0.5$, $\beta = 0.5$, $\gamma = 5$, $\delta = 1$, and:

- $\mathcal{L}_{sum\_to\_1} = \left| 1 - \sum_{i,j} k_{i,j} \right|$ encourages $k$ to sum to 1.
- $\mathcal{L}_{boundaries} = \sum_{i,j} |k_{i,j} \cdot m_{i,j}|$ penalizing non-zero values close to the boundaries. $m$ is a constant mask of weights exponentially growing with distance from the center of $k$.
- $\mathcal{L}_{sparse} = \sum_{i,j} |k_{i,j}|^{1/2}$ encourages sparsity to prevent the network from oversmoothing kernels.
- $\mathcal{L}_{center} = \left\| (x_0, y_0) - \frac{\sum_{i,j} k_{i,j} \cdot (i,j)}{\sum_{i,j} k_{i,j}} \right\|_2$ encourages $k$'s center of mass to be at the center of the kernel, where $(x_0, y_0)$ denote the center indices.

SR kernels are not only image specific, but also depend on the desired scale-factor $s$. However, there is a simple relation between SR-kernels of different scales. We are thus able to obtain a kernel for SR $\times 4$ from $G$ that was trained to downscale by $\times 2$. This is advantageous for two reasons: First, it allows extracting kernels for various scales by one run of KernelGAN. Second, it prevents downscaling the LR image too much. Small LR images downscaled by a scale factor of 4 may result in tiny images ($\times 16$ smaller than the HR image) which may not contain enough data to train on. KernelGAN is trained to estimate a kernel $k_2$ for a scale-factor of 2. It is easy to show that the kernel for a scale factor of 4, i.e. $k_4$, can be analytically calculated by requiring: $I_{LR} * k_4 \downarrow_4 = (I_{LR} * k_2 \downarrow_2) * k_2 \downarrow_2$.

It implies that $k_4 = k_2 * k_{2\_dilated}$, where $k_{2\_dilated}[n_1, n_2] = \begin{cases} k_2 \left[ \frac{n_1}{2}, \frac{n_2}{2} \right] & n_1, n_2 \text{ even} \\ 0 & \text{else} \end{cases}$

For a mathematical proof of the above derivation, as well as an ablation study of the various kernel constraints – see our project website.

## 5 Experiments and results

We evaluated our method on real LR images, as well as on 'non-ideal' synthetically generated LR images with ground-truth (both ground-truth HR images, as well as the true SR-kernels).

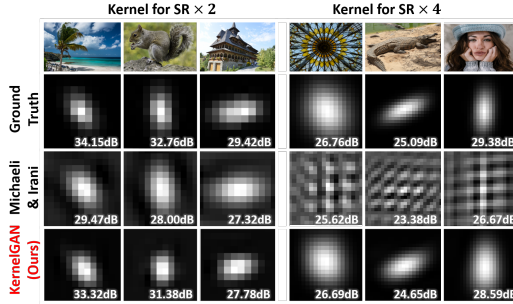

Figure 5: **SR kernel estimation**: *We compare ground truth kernel, Michaeli and Irani [24] and KernelGAN (ours). PSNR of ZSSR [30], when supplied with those kernels, is noted in the bottom right of each kernel, emphasizing the significance of kernel estimation accuracy. Our method outperforms [24] in SR performance (as well as in visual similarity to ground truth SR kernel)*

| | Method | ×2 | ×4 |
|---|---|---|---|
| **Type 1**: SotA SR algorithms (trained on bicubically downscaled images) | Bicubic Interpolation | 28.731 / 0.8040 | 25.330 / 0.6795 |
| | Bicubic kernel + ZSSR [30] | 29.102 / 0.8215 | 25.605 / 0.6911 |
| | EDSRplus [21] | 29.172 / 0.8216 | 25.638 / 0.6928 |
| | RCANplus [38] | 29.198 / 0.8223 | 25.659 / 0.6936 |
| **Type 2**: Blind-SR NTIRE'18 [31] winners | PDN [34] - 1st in NTIRE track4 | - | **26.340** / 0.7190 |
| | WDSR [36] - 1st in NTIRE track2 | - | 21.546 / 0.6841 |
| | WDSR [36] - 1st in NTIRE track3 | - | 21.539 / 0.7016 |
| | WDSR [36] - 2nd in NTIRE track4 | - | 25.636 / 0.7144 |
| **Type 3**: kernel estimation + non Blind-SR algorithm | Michaeli & Irani [24] + SRMD [37] | 25.511 / 0.8083 | 23.335 / 0.6530 |
| | Michaeli & Irani [24] + ZSSR [30] | 29.368 / 0.8370 | 26.080 / 0.7138 |
| | **KernelGAN (Ours) + SRMD [37]** | **29.565** / **0.8564** | 25.711 / **0.7265** |
| | **KernelGAN (Ours) + ZSSR [30]** | **30.363** / **0.8669** | **26.810** / **0.7316** |
| **Type 4**: Upper bound | Ground-truth kernel + SRMD [37] | 31.962 / 0.8955 | 27.375 / 0.7655 |
| | Ground-truth kernel + ZSSR [30] | 32.436 / 0.8992 | 27.527 / 0.7446 |

Table 1: *SotA SR performance (PSNR(dB) / SSIM) on 100 images of DIV2KRK (sec. 5.2) . **Red** indicates the best performance, **blue** indicates second best.*

## 5.1 Evaluation Method

The performance of our method is analyzed in 2 ways: *Kernel estimation* accuracy and *SR performance*. The latter is done both visually (see fig. 1a and supplementary material), and empirically on the synthetic dataset analyzing PSNR and SSIM measurements (Table 1). Evaluation is done using the script provided by [19] and used by many works, including [30, 20, 21]. For evaluation of the kernel estimation we chose two non-blind SR algorithms that accept a SR-kernel as input [30, 37], provide them with different kernels (bicubic, ground truth SR-kernel, ours and [30]) and compared their SR performance. We present four types (categories) of algorithms for the analysis:

- *Type 1* includes the non-blind SotA SR methods trained on bicubically downscaled images.
- *Type 2* are the winners of the NTIRE'2018 Blind-SR challenge [31].
- *Type 3* consists of combinations of 2 methods for SR-kernel estimation, [24] and ours, and 2 non-blind SR methods, [30, 37], that regard the estimated kernel as input. This combination is itself, **a full Blind-SR algorithm**.
- *Type 4*, is again the combination above, only with the use of the ground truth SR-kernel, thus providing an upper bound to *Type 3*.

When providing a kernel to ZSSR [30] (whether ours, [24]'s, ground-truth kernel), we provide both the ×2 and ×4 kernels, in order to exploit the gradual process of ZSSR (using 2 sequential SR steps).

We compare our kernel estimation to Michaeli & Irani [24] which, to the best of our knowledge, is the leading method for SR-kernel estimation from a single image.

## 5.2 Dataset for Blind-SR

There is no adequate dataset for quantitatively evaluating Blind-SR. The data used in the NTIRE'2018 Blind-SR challenge [31] has an inherent problem; it suffers from sub-pixel misalignments between the LR image and the HR ground truth, resulting in subpixel misalignments between the reconstructed SR images and the HR ground truth images. Such small misalignments are known to prefer (i.e. give lower error to) blurry images over sharp ones. A different benchmark suggested by [4] is not yet available, and is only restricted to 3 SR blur kernels. As a result, we generated a new Blind-SR benchmark, referred to as *DIV2KRK* (DIV2K random kernel).

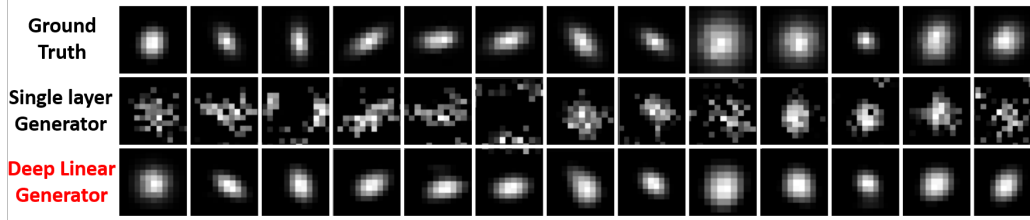

Figure 6: **Depth is of the essence for the linear network**: *The deep linear generator outperforms the single layer version by 3.8dB and 1.6dB for ×2 and ×4 over DIV2KRK. Kernel examples emphasize superiority of the deep linear network.*

Using the validation set (100 images) from the widely used DIV2K [1] dataset, we blurred and sub-sampled each image with a **different, randomly generated kernel**. Kernels were 11x11 anisotropic gaussians with random lengths $\lambda_1, \lambda_2 \sim \mathcal{U}(0.6, 5)$ independently distributed for each axis, rotated by a random angle $\theta \sim \mathcal{U}[-\pi, \pi]$. To deviate from a regular gaussian, we further apply uniform multiplicative noise (up to 25% of each pixel value of the kernel) and normalize it to sum to one. See Figs. 5 and 6 for a few such examples. Data and reproduction code are available on project website.

### 5.3 Results

**Our method together with ZSSR [30] outperforms SotA SR results visually and numerically by a large margin of 1dB and 0.47dB for scales ×2 and ×4 respectively.** When the kernel deviates from bicubic, as in *DIV2KRK* and real images, *Type 1*, SotA SR, tends to produce blurry results (often comparable to naive bicubic interpolation), and highlights undesired artifacts (e.g. JPEG compression), such examples are shown in Fig. 1a. *Type 2*, i.e. SotA Blind-SR methods, produce significantly better quantitative results, than *Type 1*. Although visually (as shown in Fig. 1b), tend to over-smooth patterns and over-sharpen edges in an artificial and graphical manner.

Our SR-kernel estimation, plugged into ZSSR [30], shows superior results over both types by a large margin of **1.1dB, 0.47dB** for scales ×2, ×4 respectively, and visually recovering small details from the LR image as shown in Fig 1. Although, our kernel together with [37] did not perform as well for scale ×4, ranking lower than [34] in PSNR (but higher in SSIM).

Regarding SR-kernel estimation, we analyze *Type 3*. When fixing a SR algorithm, our kernel estimation outperforms [24] visually, as seen in Fig. 1b, and quantitatively by **1dB, 0.73dB** when plugged in [30] and **4dB, 2.3dB** when plugged in [37] for scales ×2, ×4 respectively.

The superiority of *Type 3* and *Type 4* (as shown in Table 1) shows empirically the importance of performing SR with regard to the image specific SR-kernel. Moreover, using a SR algorithm with different SR-kernels leads to significantly different results, demonstrating the importance of the SR-kernel accuracy. This is also shown in Fig. 5 with a large difference in PSNR while small (but visible) difference in the estimated kernel. For more visual results+comparisons see project website.

**Run-time:** Network training is done during test time. There is no actual inference step since the trained $G$ implicitly contains the resulting SR-kernel. Runtime is *61 or 102 seconds* per image on a single *Tesla V-100 or Tesla K-80* GPU, respectively. Runtime is independent both of image size and scale factor, since the GAN analyzes patches rather than the whole image. Since the same kernel applies to the entire image, analyzing a fixed number of crops suffices to estimate the kernel. Fixed sized image crops (64×64 to $G$, and accordingly 32×32 to D) are randomly selected *with probability proportional to their gradient content*. KernelGAN first estimates the ×2 SR-kernel, and then derives analytically kernels for other scales (e.g., ×4). For comparison, [24] runs 45 minutes per image on our *DIV2KRK* and time consumption grows with image size.

## 6 Conclusion

We present significant progress towards real-world SR (i.e., Blind-SR), by estimating an image-specific SR-kernel based on the LR image alone. This is done via an image-specific internal-GAN, which trains solely on the LR input image, and learns its internal distribution of patches without requiring any prior examples. **This gives rise to true SR "in the wild".** We show both visually and quantitatively that when our kernel estimation is provided to existing off-the-shelf non-blind SR algorithms, it leads to SotA SR results, by a large margin.

## Footnotes

[1] Project funded by the European Research Council (ERC) under the Horizon 2020 research & innovation program (grant No. 788535)

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
