[Supplementary Material]

## Appendix– Deriving the SR-kernel for SR×4 from the SR-kernel for SR×2

We will show that the kernel for a scale factor of 4, i.e. $k_4$, can be analytically calculated from $k_2$.

**Claim**:
Given a kernel $k_2$, let $k_4$ be defined by:

$I_{LR} * k_4 \downarrow_4 = (I_{LR} * k_2 \downarrow_2) * k_2 \downarrow_2.$

Then, $k_4 = k_2 * k_{2\_dilated}$, where $k_{2\_dilated}[n_1, n_2] = \begin{cases} k_2 \left[ \frac{n_1}{2}, \frac{n_2}{2} \right] & n_1, n_2 \text{ even} \\ 0 & \text{else} \end{cases}$

**Proof**:
For simplicity, we will assume 1D signals, but this can be easily generalized to 2D.

Let us define: $x := I_{LR} * k_2$.

$$\left\{ (I_{LR} * k_2 \downarrow_2) * k_2 \right\} [n] = \left\{ k_2 * x[2 \cdot] \right\} [n] = \sum_{l=-\infty}^{\infty} \left( k_2[n - l] \cdot x[2l] \right)$$

We can substitute $m := 2l$ and use Kronecker Delta Comb to enforce $m$ to be even;

$$= \sum_{l=-\infty}^{\infty} \left( \left( \sum_{m=-\infty}^{\infty} \delta[2l - m] \right) \cdot k_2[n - \frac{m}{2}] \cdot x[m] \right)$$

Assuming both signals are of finite energy, changing the summation order is allowed. Adding $2n$ does not change the result of the Kronecker Delta Comb;

$$= \sum_{m=-\infty}^{\infty} \left( \left( \sum_{l=-\infty}^{\infty} \delta[2l - m + 2n] \cdot k_2 \left[ \frac{1}{2}(2n - m) \right] \right) \cdot x[m] \right)$$

This formulates a convolution;

$$= \left\{ \left( \sum_{l=-\infty}^{\infty} \delta[\cdot + 2l] \cdot k_2 \left[ \frac{\cdot}{2} \right] \right) * x \right\} [2n]$$

The Kronecker Delta Comb zeroizes for odd inputs, we therefore obtain $k_{2\_dilated}$;

$$= \left\{ k_{2\_dilated} * x \right\} [2n]$$

Plug in $x := I_{LR} * k_2$ and subsample both sides by $\downarrow_2$

$(I_{LR} * k_2 \downarrow_2) * k_2 \downarrow_2 = I_{LR} * (k_2 * k_{2\_dilated}) \downarrow_4$ $\qquad \square$