[Reviews · NeurIPS 2019]

Reviewer 1



This paper introduces a blind super-resolution technique, i.e. a method allowing to increase the resolution of an image without knowing the downscaling kernel. Clarity : clarity is fairly good. It's just that sometimes some statements are raising questions which are answered later in the paper. It would be better to warn the reader that explanations are coming next. originality : the proposed GAN allowing to estimate the SR kernel is new and tailored for the blind SR problem. quality : the authors have rigorously presented their approach. The paper is technically sound. The only issue I see is that the performance discrepancies reported in the experimental section should be proved to be statistically significant. significance : the results are significant (see previous comments on the contributions) Remarks : line 101 : The benefits of using a D-map as compared to a pixelwise output is not explained in details. Line 117 : having a 7x7 receptive field is also possible because in other layers, the authors use 1x1 convolutions. This should be outlined in the text. Line 130: the authors explain that using a single layer generator does not work because optimization is entwined with that of the discriminator which is non-convex. It should be mentioned here that an « over-parametrized » learned G can be anyway compacted to single layer kernel as will be done in 4.2. Table 1 : please provide evidence that the reported PSNR discrepancies are statistically significant. I do not understand what synthetic dataset is used here (It is explained in the next subsection). Reproducibility : data and code will be made available but when or under what circumstances ? Runtime : I don’t understand the comment that «  runtime is independent of image size ». If the image is larger, then there are surely more patches from which to learn and thus training epochs are probably longer. UPDATE AFTER REBUTTAL: I thank the authors for the relevant feedback they provided. Concerning statistical significance, I agree that the reported indicators in the initial submission are indeed the usual practice in the SR (or image processing) literature, but I think the authors would also agree with me that this is a bad practice. I also agree that empirical standard deviation is not very informative because what we would like to know is how "concentrated" are probabilities around the empirical mean. I suggest bootstrap confidence intervals. The figure 2 of the feedback is also interesting. About the Dmap, why is it computationally more efficient ? Is there a parallelization to be exploited ? I am confident that the authors can easily address the other suggestions I made. I maintain my score because it is already pretty high.

Reviewer 2



This paper addresses single image SR problem based on zero-shot learning. Unlike conventional learning-based SR methods which assume known SR kernel and utilize external database to train the network during training phase, this work assumes unknown SR kernel, and thus estimates the kernel in a blind manner at test time. Then, the estimated kernels are integrated with conventional SR methods (e.g., [29]), and improves restoration quality by a large margin with the aid of the accurately estimated SR kernel. Overall, this work is a natural extension of previous work [23] using learning techniques, and the manuscript is well organized and easy to read. Here are some questions and minor comments. a. Discriminator D is a binary classifier in this task, but renders a 2D map rather than a single value. Is there any specific reason to employ this network architecture? b. I believe the proposed method can estimate SR kernel even when the input images are down-scaled by bicubic interpolation. From this view (assume GT kernel is bi-cubic), it would be great if the authors provide quantitative comparison results with conventional datasets (e.g., set5, set14, ...) c. Isn’t it possible to train the SR network with train datasets whose LR images are down-scaled with various SR kernels as well as bi-cubic kernels? (i.e., SR kernels might be generated by modifying Levin et al’s blur kernel generation technique) d..How do you handle the estimated SR kernel to integrate with ZSSR[29] which is implemented on coarse-to-fine (multi-scale) manner?

Reviewer 3



This paper investigate kernel estimation method for blind super-resolution. I think this paper has the following strong points: 1) The idea of using distribution of small patches to guide the training of kernel estimator is interesting. 2) The authors proposed to use deep linear networks for estimating the kernel, and provided a good justification of using such a special network architecture. 3) The authors have used some prior information to regularize the training of kernel estimator. 4) The proposed algorithm achieved good results. Despite the above strong points, I think this paper also has the following drawbacks: 1) Lack of experimental justification. Although the authors have discussed the reason of their design choice, but it will be better if the authors could provide experimental results to show the effect of kernel constraints, kernel estimator with/without activation function ... 2) More implementation details should be provided to make this work reproducable. For example, training details, hyper-parameters ... I think there must be many tricks in selecting patches for training the discriminator. 3) is there a non-negative constraint for the estimated kernel?

[Author Response · NeurIPS 2019]

# REBUTTAL: Blind Super-Resolution Kernel Estimation using an Internal-GAN

We would like to thank the reviewers for their comments. Below are our answers to the main questions/concerns.

**R1: How can runtime be independent of image size?** The number of training iterations is predetermined and independent of image size. At each iteration a fixed-size image crop is selected (see below), and all patches within that crop are analyzed. Since the same downscaling kernel applies to the entire image, it is unnecessary to analyze all image patches (although this is likely to happen due to large number of iterations and crop size). We will add a clarification.

**R3: How are crops selected?** Thanks, we forgot to mention this. Fixed sized image crops ($64\times64$ to $G$, and accordingly $32\times32$ to D) are randomly selected with probability proportional to their mean gradient. We will add this.

**R1+R2: Why 2D map and not a single scalar output?** D provides a scalar score for every patch it analyzes, representing the probability of it belonging to the learned patch distribution. Rendering a 2D map (D-map) of all these scalars *at once* (for all the patches in the crop), in a single convolutional pass, is computationally more efficient, as proposed by [15,28] (one scalar per patch, kept in the D-map at the center pixel location of each patch). In response to R1's question, this is in fact a pixelwise output.

**R2: Performance of KernelGAN on Bicubicly downscaled images:** To verify R2's hypothesis, we ran KernelGAN for SR$\times4$ on 100 bicubicly downscaled images (DIV2K). We applied ZSSR, once with our *estimated* kernel, and once with the GT (Ground-Truth) bicubic kernel. The resulting PSNR/SSIM were 28.65dB/0.795 vs. 28.73dB/0.796, respectively. See Fig.A to view a sample of recovered kernels. We Will add this to the paper or Supp-Material. It is important to distinguish between the *Blind-SR* task and the easier *non-blind SR* task, where the GT kernel is known. External networks further train exhaustively on a large dataset of images, all downscaled with this *single* GT kernel.

**R2: Why not train SR networks on multiple degradations?** In fact, [36] reported unsuccessful experiments with this exact approach (see Chapter 3.5 of [36]: "Why not learn a blind model"). There are combinatorially many possible kernels, and each image has its own unique kernel. If a network is trained on a certain collection of kernels (e.g., random Gaussian kernels), it will be restricted to those types, and is unlikely to generalize to kernels which significantly deviate from those. In contrast, since each image has hundreds of thousands of patches, all sharing one kernel, the *unsupervised* KernelGAN (which trains on the *image-specific* patch distribution and kernel) can handle new never-before-seen kernels.

**R2: Integration with ZSSR's coarse-to-fine implementation:** We perform SR$\times2$ in a single ZSSR step, and SR$\times4$ in 2 coarse-to-fine ZSSR steps (by supplying ZSSR with both the $\times2$ and $\times4$ kernels). Note that for fair comparison, we used the same coarse-to-fine configuration when incorporating [23] into ZSSR. We will add a clarification to paper.

**R3: Add ablation study of kernel constrains:** To address R3's request, we empirically evaluated the effect of omitting each kernel constraint on 100 images (the DIV2KRK dataset). See table in Fig.C (will be added to paper/Supp-Material).

**R3: Is there a non negative constraint on the kernel?** No, the network does not impose any non-negative constraint on the kernel. Note that some kernels may include negative values (as noted by [23]) – e.g., the bicubic kernel.

**R1: Statistical significance:** We report average PSNR/SSIM on the dataset, which is the common practice in all SR papers/challenges (e.g., see NTIRE challenge report [30]). A global $std$ value per-se will not suffice, as in some images the PSNR variation among different SR methods is inherently very low, while in others it is very high. To address R1's valid concern, we provide in Fig.B the *percent* of images (out of 100) where our method (green) outperformed other methods (a very large percent). We believe this measure is a more statistically informative in the context of SR.

**R1+R3: Missing code, data and implementation details:** Code and data will be made publicly available (Github) upon acceptance. Fig.D contains implementation details. We will expand on these in the paper or Supp-Material.

**R1: Clarifications too late in text (1x1 convolutions, overparameterized G, etc.):** We'll edit and clarify accordingly.

**Fig. A:** Bicbuic Downscaling — GT-Bicubic kernel — KernelGAN estimations

**Fig. B:** Statistical Significance (SRx4)

Bicubic Interp, Bicubic+ZSSR, [23]+ZSSR, [23]+SRMD, EDSR+, RCAN+, PDN, WDSR track2, WDSR track3, WDSR track4 — NTIRE Winners

- KernelGAN better (green)
- KernelGAN worse (red)

**Fig. C:** Constraints Ablation Study

| Constraints a | SR X2 | SR X4 |
|---|---|---|
| No Sparseness | 30.32/0.862 | 26.60/0.727 |
| No Boundaries | 30.37/0.863 | 26.66/0.729 |
| No Sum to 1 | 29.82/0.851 | 26.58/0.724 |
| No Centralization | 30.43/0.865 | 26.78/0.731 |
| w/o any constraints | 29.62/0.850 | 26.32/0.721 |
| All constraints (KernelGAN) | **30.51/0.867** | **26.81/0.732** |

**Fig. D:** Implementation Details

Architecture - Fig.3,4 (in paper)
Crop size - Fig.3,4 (in paper)
Learning rate G = D = $2e^{-4}$
Learn rate update: $^{\times0.1}/_{750\ iters}$
# of iterations = 3,000
Iteration ratio: $^{G}/_{D} = ^{1}/_{1}$
Adam Optimizer ($\beta_1$=0.5, $\beta_2$=0.999)
Kernel size: $s.f.\times 2 \rightarrow 13 \times 13$
$s.f.\times 4 \rightarrow 25 \times 25$
Constraints weights:
  Sparseness=5    Boundaries=0.5
  Sum to 1=0.5    Centralization=1

*Fig. A. KernelGAN estimations for bicubicly downscaled images. Fig. B. Superiority percentage of our KernelGAN+ZSSR over each method mentioned in Table 1 of paper. Fig. C. Ablation study of the effect of different kernel constraints (PSNR(dB) / SSIM reported over 100 DIV2KRK images for SR$\times2$ and SR$\times4$). Fig. D. Implementation and hyper-parameters details.*

[Meta-Review · NeurIPS 2019]

The paper proposes a method for blind super-resolutions by estimating the kernel with a GAN. The method is based on zero-shot learning: it assumes unknown SR kernel, and thus estimates the kernel in a blind manner at test time. The method improves restoration quality by a large margin with the aid of the accurately estimated SR kernel. The paper is well written. Reviewers agreed since the beginning on the acceptance and are satisfied by the rebuttal. Thus the area chair agrees with an acceptance.